# Comparative Analysis of Antioxidant Accumulation under Cold Acclimation, Deacclimation and Reacclimation in Winter Wheat

**DOI:** 10.3390/plants11212818

**Published:** 2022-10-23

**Authors:** Gabija Vaitkevičiūtė, Andrius Aleliūnas, Yves Gibon, Rita Armonienė

**Affiliations:** 1Lithuanian Research Centre for Agriculture and Forestry, Institute of Agriculture, Instituto al. 1, LT-58344 Kėdainiai, Lithuania; 2UMR 1332 Biologie du Fruit et Pathologie (BFP), INRAE, Université de Bordeaux, 33882 Bordeaux, France

**Keywords:** abiotic stress, ascorbate, climate change, freezing tolerance, glutathione, low-temperature stress, *Triticum aestivum* L., winter hardiness

## Abstract

Low temperature during cold acclimation (CA) leads to the accumulation of detrimental reactive oxygen species (ROS) in plant tissues, which are scavenged by antioxidants, such as ascorbate and glutathione. However, there is a lack of studies examining the dynamics of antioxidants throughout CA, deacclimation (DEA), and reacclimation (REA) in winter wheat. Six winter wheat genotypes were selected to assess the effect of CA, DEA, and REA on the concentrations of ascorbate and glutathione in leaf and crown tissues under two CA temperature treatments. Higher levels of total, reduced, and oxidised ascorbate were determined in leaves, whereas crowns accumulated higher concentrations of nicotinamide adenine dinucleotide (NAD^+^) after CA, DEA, and REA. Constant low temperature (CLT) during CA led to higher contents of ascorbate and glutathione in both tissues at all stages of acclimation, in comparison with prolonged higher low temperature (PHLT). The concentrations of antioxidants increased after CA, tended to decrease after DEA, and returned to CA levels after REA. Significant positive correlations between freezing tolerance (FT) and antioxidants were only determined under the CA at CLT treatment, thus, affirming the negative effect of PHLT during CA on the FT of winter wheat.

## 1. Introduction

Reactive oxygen species (ROS) are produced by all aerobic life forms. They include oxygen radicals and derivatives, such as superoxide (O_2_^−^), hydroxyl (OH^⋅^), and hydrogen peroxide (H_2_O_2_) [1]. At small concentrations, they act as signalling molecules; however, excessive oxidative stress can lead to cell death. The negative effects of ROS include protein oxidation, lipid peroxidation, and DNA damage [2]. Plants are known to be especially effective at ROS production and scavenging due their photosynthetic nature. Chloroplasts, mitochondria, and peroxisomes produce the highest concentrations of ROS; therefore, the balance between ROS production and scavenging is important to ensure the normal functioning and survival of plant cells [2]. Moreover, abiotic stresses, such as high salinity [3], drought [4], heavy metals [5], and low temperature [6] had been shown to result in excessive ROS accumulation. Consequently, plants have evolved two types of ROS scavenging systems—enzymatic and non-enzymatic. As reviewed by Mittler et al. (2004), enzymatic ROS scavengers include, among others, ascorbate peroxidase (APX) (EC 1.11.1.11), superoxide dismutase (SOD) (EC 1.15.1.1), glutathione peroxidase (GPX) (EC 1.11.1.9), and catalase (CAT) (EC 1.11.1.6), whereas non-enzymatic ROS scavengers are the reduced forms of antioxidants, such as ascorbate (ASC, also known as vitamin C) and glutathione (GSH) [7]. These antioxidants are central in plant ROS scavenging, and they comprise the ascorbate-glutathione (ASC-GSH) cycle [8]. The main function of the ASC-GSH cycle is to reduce excess H_2_O_2_ into H_2_O and thus protect the cells from oxidative damage. The enzyme APX uses ASC as an electron donor to reduce H_2_O_2_, and as a result, ASC is oxidised to dehydroascorbate (DHA) [9,10]. Subsequently, DHA molecules are non-enzymatically reduced back into ASC by GSH. During this process GSH is oxidised into glutathione disulphide (GSSG) [9]. GSSG is then reverted into GSH by the reduced nicotinamide adenine dinucleotide phosphate (NADPH)-dependant enzyme glutathione reductase (GR) (EC 1.8.1.7). During the GSSG reduction reaction, GR produces 2 GSH molecules and one oxidised nicotinamide adenine dinucleotide phosphate (NADP^+^) molecule [9].

Low temperatures, ranging from 0 to 10 °C, and reduced light conditions in autumn induce a cascade of changes at the transcriptome and metabolome levels in overwintering crops, leading to the process known as cold acclimation (CA) or winter hardening [11,12,13,14]. To reach the highest level of freezing tolerance (FT), CA in winter wheat (*Triticum aestivum* L.) takes between 28 and 56 days, depending on the genotype. Fully acclimated winter wheat can survive negative temperatures as low as −20 °C [11,15]. Following CA, rising temperatures result in deacclimation (DEA), after which winter type crops lose their FT and can begin the transition from vegetative into the reproductive stage. However, some plants possess the advantageous ability to reacclimate (REA) and regain FT when they are repeatedly exposed to decreasing temperatures [16,17]. ROS play a significant role in the acclimation process, as they can both act as signalling molecules, and as harmful agents [18]. For example, H_2_O_2_ down-regulates the expression of a *CBF3* gene in *Arabidopsis* [19]. This gene is a component in the ICE-CBF-COR pathway, which is the main CA signalling cascade [20]. Therefore, winter type crops have evolved specific response mechanisms to scavenge the excess of ROS molecules and ensure survival under low negative temperatures. These mechanisms involve increased transcription of antioxidant-related genes, accumulation of antioxidants, and higher antioxidant-related enzyme activity [21,22,23].

There is great concern that the autumn season in temperate regions, where winter wheat is the major cereal crop, will become increasingly longer and warmer due to global climate change. Prolonged warmer autumns will lead to delayed and insufficient CA under shortened photoperiod and decreased light intensity conditions. Furthermore, temperature fluctuations at wintertime will induce premature DEA, ultimately reducing the yield and quality of grains [16,24,25,26]. Nevertheless, the processes of DEA and REA are not yet fully understood and require further investigation [16,27]. Our recent study showed that prolonged higher low temperature during CA significantly affects metabolite accumulation in winter wheat throughout CA, DEA, and REA [28]. However, there are insufficient studies on the dynamics of specific antioxidants in winter wheat throughout these stages of acclimation. The aim of this study is to evaluate the effect of CA, DEA, and REA on the concentrations of total, reduced, and oxidised ascorbate and glutathione in leaf and crown tissues of six winter wheat genotypes under two CA temperature treatments, and to determine the relationships between these antioxidants and FT.

## 2. Results

### 2.1. Principal Component Analyses of Antioxidants

The antioxidant accumulation data were projected on principal component analysis (PCA) dimensions 1 and 2, and separations according to tissue, stage of acclimation, treatment, and genotype were assessed (Figure 1). Dimension 1 accounted for 44.2% of total variation and set oxidised nicotinamide adenine dinucleotide (NAD^+^) apart from the remaining antioxidant variables (Figure 1A). Dimension 2 explained 26.5% of the observed variation, which resulted in a clear distinction between ascorbate and glutathione. Moreover, NADP^+^ showed a positive correlation with ascorbate variables. The PCA clearly distinguished two groups of the measured antioxidants by tissue (Figure 1B). Three clusters of non-acclimated control (CTRL), constant low temperature (CLT), and prolonged higher low temperature (PHLT) treatments were observed (Figure 1C). The distinction between stages of acclimation was weaker, with the CTRL stage group being most divergent (Figure 1D). Genotype had no significant effect on antioxidant separation (Appendix A).

### 2.2. Assessment of Ascorbate Accumulation

Significant differences of total ascorbate accumulation were found between both tissues (*p* < 0.001) and treatments (*p* < 0.01) at the stages of CA, DEA, and REA (Figure 2A, Appendix A). The levels of total ascorbate were the highest in leaves, subjected to the CLT treatment. The concentration of total ascorbate in leaves was lowest in non-acclimated winter wheat (CTRL) and increased 3.1 times after 49 days of CA at 2 °C in the CLT treatment. DEA resulted in decreased levels of total ascorbate in leaves. Total ascorbate content remained identical to DEA after REA. In contrast, the PHLT treatment resulted in a 1.6-fold higher concentration of total ascorbate after CA in comparison with the CTRL time-point. The increased levels of ascorbate remained stable throughout DEA and REA.

The concentrations of ASC and DHA were likewise higher in leaf tissue and showed a tendency to remain elevated under the CLT treatment (Figure 2B,C; Appendix A), in comparison to the PHLT treatment. The accumulation of ASC and DHA increased significantly after CA under the CLT treatment and decreased after DEA. Markedly, the ASC/DHA ratio was the highest in crown tissue under the CLT treatment after CA and DEA, compared to leaf tissue (Figure 2D; Appendix A).

### 2.3. Determination of Glutathione Concentrations

Higher concentrations of total glutathione content, GSH, and GSSG were observed in leaf tissue under the CLT treatment at the stages of CA, DEA, and REA, as compared to the PHLT treatment (Figure 3, Appendix A). A similar tendency was found in crown tissue after CA and REA. The levels of total glutathione, GSH, and GSH/GSSG ratio increased significantly (*p* < 0.05) in both tissues and treatment groups after CA. Following DEA, the accumulation of total glutathione and GSH in leaves significantly decreased and recovered to CA levels after REA in both treatment groups.

The concentrations of GSSG were significantly (*p* < 0.05) higher in leaf tissue after CA and DEA in both treatments, as compared to crown tissue (Figure 3D). Moreover, the GSH/GSSG ratio was lower in both leaf and crown tissues after CA under the CLT treatment, compared to the PHLT treatment.

### 2.4. Assessment of NAD^+^ and NADP^+^ Accumulation

Crowns tended to contain higher concentrations of NAD^+^ than leaves throughout the entire experiment in both acclimation treatment groups (Figure 4A, Appendix A). Furthermore, the levels of crown NAD^+^ were significantly (*p* < 0.01) higher under the CLT treatment in comparison to the PHLT treatment after CA, DEA, and REA (Appendix A). CA resulted in increased leaf NAD^+^ content in the CLT treatment in comparison to the CTRL sampling point. The concentration decreased to CTRL levels after DEA. Following REA, it once again increased to the levels, observed after CA (Figure 3A).

Leaves showed a tendency to contain higher concentrations of NADP^+^ than crowns (*p* < 0.05) (Figure 4B, Appendix A). However, after DEA and REA in the CLT treatment, the levels were higher in crown tissue instead (*p* < 0.01). CA resulted in increased accumulation of NADP^+^ in leaves under both treatments compared to CTRL. DEA and REA lead to significantly decreased levels of NADP^+^ in leaves under the CLT treatment, in comparison with CA. Under the PHLT treatment, the concentration of NADP^+^ in leaves remained stable after DEA and decreased to CTRL levels after REA (Figure 4B).

### 2.5. Correlations between Antioxidants and Freezing Tolerance

LT_30_ (temperature, at which 30% of plants die) of six winter wheat genotypes, determined in our previous study [28], was used as an indicator of FT in winter wheat (Appendix A). The significance of relationships between antioxidants, measured in leaf and crown tissue under two separate treatments, and FT throughout CA, DEA, and REA was determined (Table 1). The CLT treatment resulted in significant correlations, whereas no significant correlations were yielded by the PHLT treatment. A moderate positive (r ≥ 0.4) correlation was found between FT and ASC/DHA ratio, and strong positive (r ≥ 0.6) correlations [29] were shown with total ascorbate, ASC, total glutathione, GSH, and NADP^+^ in leaf tissue. In the crown tissue, moderate positive correlations were detected between FT and total glutathione, GSH and NAD^+^, and strong positive correlations were found with total ascorbate and DHA.

## 3. Discussion

### 3.1. Antioxidant Accumulation Patterns Are Tissue-Specific

Photosynthetic tissues, such as shoots and leaves, have previously been shown to contain higher levels of ascorbate in comparison to non-photosynthetic tissues, e.g., roots [30]. Evidently, exposure to light induces the development of chloroplasts, followed by the increased transcription of ascorbate biosynthesis-related genes, as well as heightened concentrations of total ascorbate in *Arabidopsis* root cells [31]. According to our study, leaves accumulated higher concentrations of total ascorbate, ASC, and DHA under both treatment groups at the stages of CA, DEA, and REA compared to crowns. Moreover, under the PHLT treatment, leaves tended to contain higher levels of NADP^+^, which showed a positive correlation with ascorbate variables in the PCA biplot (Figure 1B). NADP^+^ is associated with anabolic reactions, especially photosynthesis [32], which explains the abundance of this molecule in leaf tissue, as well as its correlation with ascorbate in our study. Notably, the levels of GSSG were likewise higher after CA and DEA under both treatment groups in leaf tissue, in comparison to crown tissue. Photosynthetic reactions are known to generate an abundance of ROS molecules; however, excessive ROS production ultimately results in the inhibition of photosynthetic activity [33,34]. Therefore, effective ROS scavenging, where ascorbate and glutathione play a central role via the ASC-GSH cycle, ensure the continued survival and growth of plants under stress. The increased activity of GR, which reduces GSSG into GSH, has previously been recorded in response to heavy metals, cold, salt, heat, and drought stress [35,36,37]. Similarly, in this study, the leaves of winter wheat begin to suffer from low-temperature-induced oxidative stress during CA, as photosynthesis is limited and the reaction rates are decreased [38]. This stress is thus combated with increased antioxidant content.

Moreover, crowns consistently contained more NAD^+^ under both treatment groups throughout the entire experiment, in comparison with leaf tissue. As reviewed by Gakière et al. (2018), NAD^+^ plays an important role in redox, signalling, pathogen defence, biotic and abiotic stress response, and catabolic reactions [39]. Furthermore, NAD^+^ is used as a substrate by histone and transcription factor deacetylases, which epigenetically regulate the expression of genes [40,41]. If the above-ground section of a winter type crop suffers freezing damage, the survival of actively dividing meristematic cells within the crown region ensures the eventual regrowth of the plant [42]. Our earlier study showed higher levels of amino acids, soluble carbohydrates, hexose phosphates, citrate, and malate in the crowns, compared to the leaves [28]. Therefore, the concurrently elevated concentration of NAD^+^ further substantiates the increased metabolic activity within the crowns during CA, DEA, and REA, with the ultimate purpose of protecting the meristematic cells from freezing damage.

### 3.2. Constant Low Temperature during CA Results in a Stronger Accumulation of Antioxidants

Several recent studies have explored the negative effect of prolonged warmer CA temperature on photosynthetic activity and FT in forage grasses [24,25]. Moreover, our recently published research showed that prolonged higher low-temperature during CA results in significant metabolite profile changes, increased shoot biomass accumulation, and lower FT in winter wheat [28]. Decreased concentrations of antioxidant molecules after CA, DEA, and REA under the PHLT treatment in comparison to the CLT treatment were determined in this study. The difference between the treatment groups was constant in total ascorbate concentration in both leaf and crown tissues, especially during CA, when the concentration of total ascorbate was on average two-fold higher in CLT compared to PHLT. An identical tendency was likewise observed in total glutathione accumulation. The significantly increased ascorbate and glutathione levels under CLT, in comparison with the PHLT treatment, point to a strong response of redox homeostasis maintenance mechanisms. Moreover, the CLT treatment resulted in a higher FT of winter wheat after 49 days of CA [28]. Low temperature during CA promotes the activity of enzymes, related to the ascorbate-glutathione cycle, thus, improving ROS scavenging capacity and efficacy of photosynthesis in winter type crops [21,22]. Higher concentrations of ascorbate and glutathione have likewise been shown to improve winter wheat FT [23].

Notably, the ASC/DHA ratio was significantly lower under the CLT treatment in leaves, and the GSH/GSSG ratio was lower in both leaves and crowns after CA, as compared to the PHLT treatment. As reviewed by Foyer and Noctor (2011), the ratio of reduced to oxidised antioxidants are indicators of the antioxidative capacity of the cell [43]. Despite the CLT treatment resulting in higher total ascorbate and glutathione levels, the low-temperature stress and elevated ROS accumulation pose a significant challenge to the redox capacity of the plants. However, the lowered ratio of reduced to oxidised antioxidants does not necessarily signify poor performance, as the CLT treatment resulted in increased FT after CA in comparison with PHLT [28]. 

Furthermore, CLT yielded almost two-fold higher concentrations of NAD^+^ in the crown tissue after CA, as compared with PHLT. This difference lessened yet remained significant after DEA and REA. As discussed previously, the increased concentration of NAD^+^ in the crown region may be indicative of its role in the plant’s survival and regrowth, as it is needed for catabolic reactions [39]. The levels of NADP^+^, which is required for anabolic reactions [44], were likewise higher in crowns under the CLT treatment after CA and REA. As opposed to the two-step CA process under the PHLT treatment, the stable CA process under the CLT treatment increased the catabolic and anabolic rate, as well as ROS scavenging reactions in the crowns, ultimately leading to the improved FT of winter wheat.

### 3.3. Concentrations of Antioxidants Decrease after DEA and Increase after REA

In this study, the 49 days of CA resulted in significantly increased concentrations of total ascorbate and total glutathione in both tissues and treatment groups; however, this increase was the strongest in leaf tissue under the CLT treatment. A similar tendency was observed for ASC and GSH concentrations. The levels of these antioxidants showed a decreasing trend after DEA. However, after REA, the concentration of total ascorbate remained unchanged, while the concentration of total glutathione returned to CA levels in leaves and increased even more in crowns under the CLT treatment. Pukacki and Kamińska-Rożek (2013) similarly reported a decrease in ASC and GSH concentrations in *Picea* needles after DEA, in comparison with CA [45]. A study on *Sabina* species revealed an increase in ASC and GSH concentrations, as well as heightened antioxidant enzyme activity in leaf tissue in response to decreasing temperatures during CA. However, as the temperatures began to increase and DEA occurred, the levels of ASC and GSH and the activity of these enzymes grew once again [46]. In a contrasting manner, the activity of antioxidant enzymes in winter wheat leaves have been shown to increase after CA, tended to decrease after DEA, and once again to increase after REA [47].

Although the six winter wheat genotypes showed different levels of FT, there was a lack of significant differences in antioxidant concentrations between them (Appendix A). Therefore, the data from all genotypes were pooled to carry out the statistical analyses. These similarities between the genotypes may be the result of strong evolutionary conservation of the antioxidant response mechanisms. Phylogenetic analyses of ascorbate-glutathione cycle enzymes in *Arabidopsis*, *Poaceae*, and numerous other plant species reveal strongly conserved motifs [37,48,49,50]. Moreover, plants possess multiple isoforms of antioxidant enzymes, which can have different expression patterns and activities in response to abiotic stress [35,51]. A future study could be focused on the activity of different antioxidant enzyme isoforms in leaf and crown tissues of multiple winter wheat genotypes throughout CA, DEA, and REA.

### 3.4. Antioxidants Show Correlations with Freezing Tolerance under Constant Low-Temperature Treatment

An earlier study by Pukacki and Kamińska-Rożek (2013) reported a strong positive correlation between FT and ASC content, and a moderate positive correlation between FT and GSH content in needles of *Picea* species [45]. Our results support these findings, as significant positive correlations were determined between winter wheat FT and total ascorbate, total glutathione, and GSH concentrations in both leaf and crown tissues under the CLT treatment. A higher number of stronger correlations were found in the leaf tissue than in the crown tissue; furthermore, correlations between FT and ASC, ASC/DHA ratio, and NADP^+^ were restricted to the leaves. Winter wheat possess the ability to retain higher efficacy of photosynthesis under low temperatures, in comparison with spring wheat [52], and ascorbate along with NADP^+^ both play a role in the metabolism and ROS scavenging of photosynthetic tissues [30,32]. Therefore, higher levels of antioxidants, responsible for the scavenging of ROS and the maintenance of photosynthetic activity throughout CA, can ultimately lead to improved FT.

In this study, correlations between FT and DHA and NAD^+^ were specific to the crown region. The concentrations of NAD^+^ were constantly higher in the crown tissue, and highest levels were observed in the crown tissue under the CLT treatment. As mentioned previously, NAD^+^ is a substrate for redox and catabolic reactions [39] and plays a role in the epigenetic regulation of gene expression [40,41]. NAD^+^ accumulation patterns, along with its correlation with FT shown here, strongly indicate its importance in the survival of the crown region under low negative temperatures.

No significant correlations were found with any of the antioxidant variables under the PHLT treatment conditions. The reason behind this could be the relatively lower antioxidant accumulation in both tissues under the PHLT treatment, compared to the CLT treatment. Coincidentally, the PHLT treatment resulted in significantly lowered FT after CA. The insufficient accumulation of antioxidants under the PHLT treatment is likely one of the factors, influencing the reduced FT of winter wheat after CA. These findings raise strong concern for future climate change, and its negative effect upon the survival of winter type crops.

## 4. Materials and Methods

### 4.1. Plant Material and Growth Conditions

Six commercially available varieties of winter wheat (‘KWS Ferrum’, ‘Hanswin’, ‘Nordkap’, ‘Sedula DS’, ‘SW Magnifik’ and ‘Lakaja DS’) displaying different levels of FT were chosen for this study. The seeds were imbibed and sown, and plants were grown under previously described conditions [28]. The wheat plants were subjected to two different low-temperature treatment groups of CA, with the first group consisting of constant low-temperature (CLT) at 2 °C for 49 days, and the second group consisting of prolonged higher low temperature (PHLT) at 10 °C for 28 days and 2 °C for the following 21 days. After 49 days of CA, both treatments identically included 7 days of DEA at 10 °C and 14 days of REA at 2 °C [28].

### 4.2. Sample Collection and Processing for Antioxidant Assays

Samples of winter wheat leaf and crown tissues were collected in the middle of the photoperiod. A total of 4 sampling points were chosen: prior to CA treatment on day 0, after CA on day 49, after DEA on day 56, and after REA on day 70. A pooled sample of three individual plants was used as one biological replicate. Three biological replicates were collected for every genotype, treatment group and sampling point. Samples were flash-frozen in liquid nitrogen. Fresh weight (FW) aliquots of 50 and 20 mg of leaf and crown tissues, respectively, were prepared and stored at −80 °C. 

### 4.3. Antioxidant Assays

Extractions were carried out in 500 μL of HCl (0.1 M). Total ascorbate and ASC were measured according to Stevens et al. (2006) [53] using a Thermo Multiskan Ascent microplate reader (Thermo Fisher Scientific, Waltham, MA, USA). DHA was calculated by subtracting ASC from total ascorbate. Total glutathione and GSSG, NAD^+^, and NADP^+^ were measured following the protocol by Queval and Noctor (2007) [54]. GSH was calculated according to Equation (1) [54]:(1)GSH=total glutathione−2×GSSG

Total glutathione, GSSG, NAD^+^ and NADP^+^ measurements were carried out on the MP96 microplate reader (Safas, Monaco). Assays were automated using a Microlab STAR automated liquid handling platform (Hamilton Robotics, Bonaduz, Switzerland).

### 4.4. Statistical Analyses

Statistical analyses were carried out on R v. 4.1.1 (University of Auckland, Auckland, New Zealand) [55]. As the initial statistical analyses showed a lack of significant differences in antioxidant accumulation between the six winter wheat genotypes (Appendix A), the data from all genotypes were pooled for the subsequent statistical analyses comparing tissues, treatments, and stages of acclimation. The Shapiro–Wilk test was used to assess the normality of data [56]. Non-normally distributed data were analysed by applying Wilcoxon rank-sum [57] and Kruskal–Wallis H [58] tests. LT_30_ (temperature at which 30% of plants die) was used as an indicator of FT in winter wheat [59]. LT_30_ data were collected at the CA, DEA, and REA stages of this experiment (Appendix A), and is available online along with the methodology [28]. The correlations between FT and antioxidant concentrations were assessed with the Spearman’s Rank Correlation Coefficient [60]. Principal component analysis (PCA) plots were drawn using the R package ‘factoextra’ [61].

## 5. Conclusions

Strong segregation in the accumulation of antioxidants in leaf and crown tissues after CA, DEA, and REA was identified in this study. The leaf tissue accumulated higher concentrations of total ascorbate, ASC, and DHA, whereas the crown tissue was found to contain higher concentrations of NAD^+^ throughout the entire experiment. The CLT treatment yielded higher concentrations of total ascorbate and glutathione in both tissues, in comparison to the PHLT treatment. Moreover, the concentrations of total ascorbate and glutathione increased after CA in both tissues under the CLT treatment, in comparison with the non-acclimated CTRL time-point. DEA resulted in a decrease in these antioxidants. Following REA, the levels of total ascorbate remained constant, whereas the concentrations of total glutathione returned to CA levels in leaves and increased even more in the crown region under the CLT treatment. Additionally, positive correlations were determined between FT and concentrations of total glutathione, GSH, and total ascorbate in winter wheat leaf and crown tissues under the CLT treatment. Correlations between FT and ASC/DHA ratio, and NADP^+^ were leaf-specific, whereas correlations between FT and DHA and NAD^+^ were crown-specific. The antioxidant profiles and dynamics during CA, DEA, and REA reflect the photosynthetic function of leaves, and the regenerative role of the crown region in response to low-temperature stress. Further, more profound future research, focusing on the temporal dynamics and tissue-specific activity of multiple isoforms of antioxidant enzymes in winter wheat throughout CA, DEA, and REA, is required.

## Figures and Tables

**Figure 1 plants-11-02818-f001:**
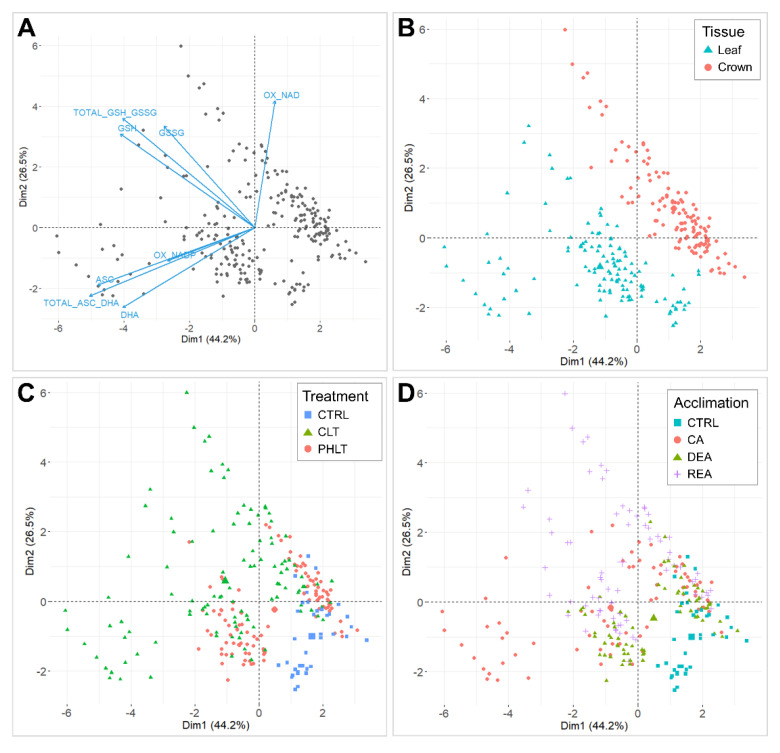
Principal component analyses (PCAs) of antioxidants, measured throughout the experiment. A biplot of individuals and variables, where positively correlated variables point towards the same direction (**A**). Biplot, grouped by tissue (**B**). Biplot, grouped by treatment (**C**). Biplot, grouped by stage of acclimation (**D**). The control (CTRL) sampling point preceded the acclimation treatments and was followed by cold acclimation (CA), deacclimation (DEA), and reacclimation (REA). CLT—constant low-temperature treatment during CA; PHLT—prolonged higher low-temperature treatment during CA.

**Figure 2 plants-11-02818-f002:**
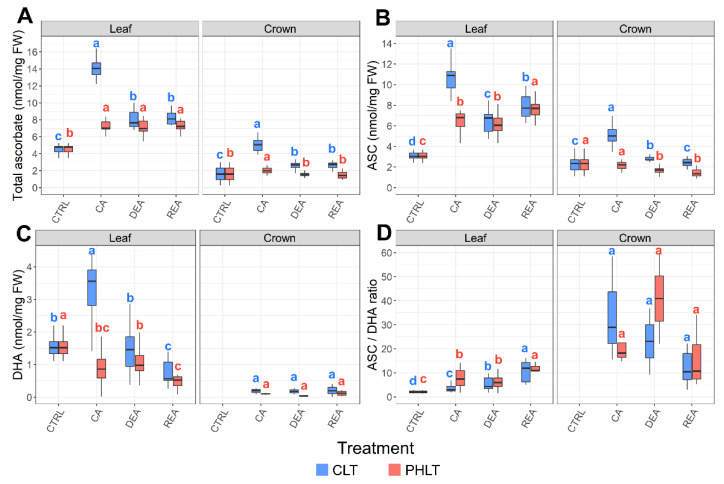
The effect of cold acclimation (CA), deacclimation (DEA), and reacclimation (REA) on ascorbate accumulation in leaf and crown tissues of winter wheat under two low-temperature treatments. The dynamics of total ascorbate (**A**), reduced ascorbate (ASC) (**B**), oxidised ascorbate (DHA) (**C**), and the ratio of reduced to oxidized ascorbate (ASC/DHA) (**D**) are provided. The letters above the boxplots indicate significant (*p* < 0.05) differences between stages of acclimation within each tissue type and treatment group. CLT—constant low-temperature treatment during CA; PHLT—prolonged higher low-temperature treatment during CA. The control (CTRL) sampling point preceded the acclimation treatments; therefore, the concentrations at this stage are identical. The concentration of DHA in crown tissue at the CTRL sampling point was below the detection threshold.

**Figure 3 plants-11-02818-f003:**
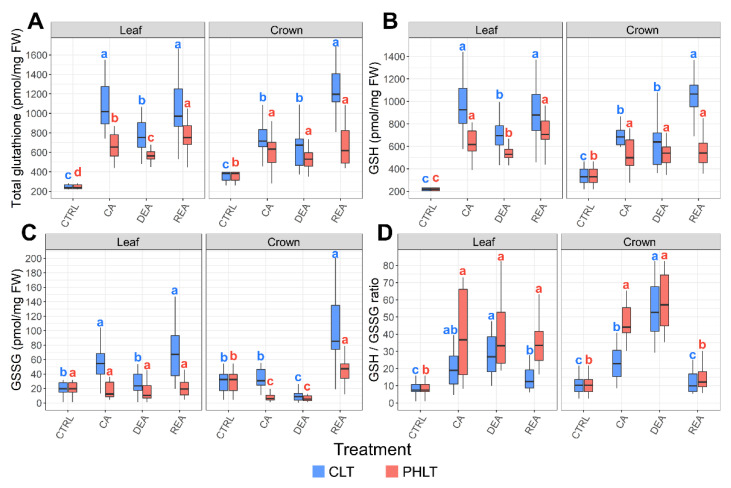
The effect of cold acclimation (CA), deacclimation (DEA), and reacclimation (REA) on glutathione accumulation in leaf and crown tissues of winter wheat under two low-temperature treatments. The dynamics of total glutathione (**A**), reduced glutathione (GSH) (**B**), oxidised glutathione (GSSG) (**C**), and the ratio of reduced to oxidized glutathione (GSH/GSSG) (**D**) are provided. The letters above the boxplots indicate significant (*p* < 0.05) differences between stages of acclimation within each tissue type and treatment group. CLT—constant low-temperature treatment during CA; PHLT—prolonged higher low-temperature treatment during CA. The control (CTRL) sampling point preceded the acclimation treatments; therefore, the concentrations at this stage are identical.

**Figure 4 plants-11-02818-f004:**
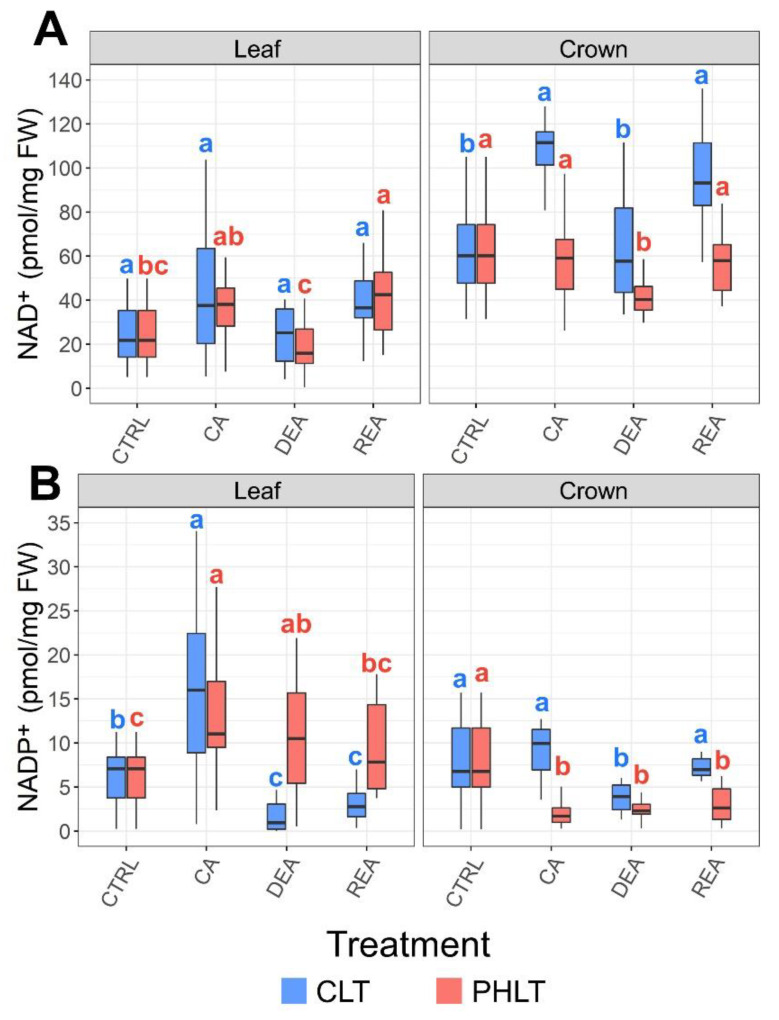
The effect of cold acclimation (CA), deacclimation (DEA), and reacclimation (REA) on accumulation of oxidised nicotinamide adenine dinucleotide (NAD^+^) (**A**) and oxidized nicotinamide adenine dinucleotide phosphate (NADP^+^) (**B**) in leaf and crown tissues of winter wheat under two low-temperature treatments. The letters above the boxplots indicate significant (*p* < 0.05) differences between stages of acclimation within each tissue type and treatment group. CLT—constant low-temperature treatment during CA; PHLT—prolonged higher low-temperature treatment during CA. The control (CTRL) sampling point preceded the acclimation treatments; therefore, the concentrations at this stage are identical.

**Table 1 plants-11-02818-t001:** Spearman coefficients (r values) of antioxidants with freezing tolerance throughout the experiment. Strong (r ≥ 0.6) and moderate (r ≥ 0.4) positive correlations are indicated by blue and light blue colours, respectively. Non-significant correlations (*p* > 0.05) are marked as “ns”.

	Leaf Tissue		Crown Tissue	
	CLT	PHLT	CLT	PHLT
Total ascorbate	0.63	ns	0.68	ns
ASC	0.63	ns	ns	ns
DHA	ns	ns	0.69	ns
ASC/DHA ratio	0.51	ns	ns	ns
Total glutathione	0.65	ns	0.57	ns
GSH	0.69	ns	0.59	ns
GSSG	ns	ns	ns	ns
GSH/GSSG ratio	ns	ns	ns	ns
NAD^+^	ns	ns	0.49	ns
NADP^+^	0.63	ns	ns	ns

## Data Availability

The data presented in this study are available in the Appendix A.

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
