# Peer review of "Comparative Analysis of Antioxidant Accumulation under Cold Acclimation, Deacclimation and Reacclimation in Winter Wheat"

_plants, 2022, doi:10.3390/plants11212818_

Round 1

Reviewer 1 Report

The MS entitled “Comparative analysis of antioxidant accumulation under cold acclimation, deacclimation and reacclimation in winter wheat” with authors Gabija Vaitkevičiūtė, Andrius Aleliūnas, Yves Gibon and Rita Armonienė is very interesting, its topic is important, and is very well written.

However, I noticed some important issues which need to be fixed,

1)      Excel files showed that 6 different winter wheat varieties were used for all biochemical measurements. The same is stated in the Materials and Method section. However, it is not clear how exactly the data from all 6 wheat varieties were included in the data presented (Figures 1-4 and Tables). Therefore, it is necessary to give more information for data collection and processing in the Materials and Method section.

2)      If all the data (from the 6 different winter wheat genotypes) were pooled, a more detailed explanation for that decision is necessary for the Discussion section. There is some discussion concerning the evolutionary conservation of the antioxidant mechanisms, but the decision to combine data obtained from different varieties is not usual, so the discussion concerning it needs to be bigger.

3)      The original results showing the data (with appropriate statistics) from all different varieties possessing different levels of freezing tolerance should stay in some form (as supplementary figures for example).

4)    If Table S1 and Table S2 possess identical data, it will be better to choose one of them.

In addition, it is necessary to include the abbreviation description in the Abstract (FT) and within the text of Figure 1.

Within the pdf file, there are the same suggestions and the text is highlighted

Author Response

The authors would like to thank the reviewers for their constructive comments and suggestions. We have carefully considered the comments and tried our best to address every one of them. 

1)      Excel files showed that 6 different winter wheat varieties were used for all biochemical measurements. The same is stated in the Materials and Method section. However, it is not clear how exactly the data from all 6 wheat varieties were included in the data presented (Figures 1-4 and Tables). Therefore, it is necessary to give more information for data collection and processing in the Materials and Method section.

A supplementary figure (Figure S1) was included to support the lack of differences in antioxidant accumulation between the 6 winter wheat genotypes. Further changes, done according to this suggestion, are described below with the lines provided.

2)      If all the data (from the 6 different winter wheat genotypes) were pooled, a more detailed explanation for that decision is necessary for the Discussion section. There is some discussion concerning the evolutionary conservation of the antioxidant mechanisms, but the decision to combine data obtained from different varieties is not usual, so the discussion concerning it needs to be bigger.

Changes, done according to this suggestion, are described below with the lines provided. The explanation was provided in the Discussion and Methods sections, and an additional supplementary figure (Figure S1) was provided.

3)      The original results showing the data (with appropriate statistics) from all different varieties possessing different levels of freezing tolerance should stay in some form (as supplementary figures for example).

A supplementary figure (Figure S2) was included to show the LT30 data in 6 winter wheat genotypes under two treatments at the stages of cold acclimation, deacclimation, and reacclimation. As this LT30 data is provided and discussed extensively in our earlier paper (Vaitkevičiūtė et al., 2022; doi: 10.3389/fpls.2022.959118), a reference is included.

4)    If Table S1 and Table S2 possess identical data, it will be better to choose one of them.

Although the tables appear to be similar, Table S1 compares the antioxidant concentrations between tissues, whereas Table S2 compares the antioxidant concentrations between treatments. Therefore, both tables should be included to fully support the figures and results, discussed in the manuscript.

Below are provided the changes, carried out according to the comments by Reviewer 1. For clarity, here we refer to the line numbers from the original manuscript.

Line 21: abbreviation was explained - “freezing tolerance” (FT).

Line 98: a supplementary figure (Figure S1) was included and referred to show the lack of differences in antioxidant concentrations between the six winter wheat genotypes.

Lines 100-103: abbreviation descriptions for “CTRL”, “CA”, “DEA”, “REA”, “CLT” and “PHLT” were included.

Line 178: a supplementary figure (Figure S2) was included and referred to show the LT30 data in 6 winter wheat genotypes.

Line 200: “which coincidentally had a positive correlation” changed to “which showed a positive correlation”.

Line 280-281: a supplementary figure (Figure S1) was included and referred to show the lack of differences in antioxidant concentrations between the six winter wheat genotypes.

Line 280-282: the section was rewritten as “Although the six winter wheat genotypes showed different levels of FT, there was a lack of significant differences in antioxidant concentrations between them (Figure S1). Therefore, the data from all genotypes was pooled to carry out the statistical analyses. These similarities between the genotypes may be the result of strong evolutionary conservation of the antioxidant response mechanisms”. The remaining paragraph (lines 282-288) provides further discussion and propose possible future research.

Line 350: the following sentence was added to clarify the process of data analysis: “As the initial statistical analyses showed a lack of significant differences in antioxidant accumulation between the six winter wheat genotypes (Figure S1), the data from all genotypes was pooled for the subsequent statistical analyses comparing tissues, treatments, and stages of acclimation”.

Line 354: a supplementary figure (Figure S2) was included and referred to show the LT30 data in 6 winter wheat genotypes.

Reviewer 2 Report

The present manuscript aims to elucidate the dynamics of the ASC-GSH system in winter wheat under cold acclimation, deacclimation, and reacclimation. The relationships between these antioxidants and freezing tolerance were investigated by determining the concentrations of total, reduced and oxidized ascorbate and glutathione in leaf and crown tissues of six winter wheat genotypes under two low-temperature acclimation treatments, as well as NAD+ and NADP+ accumulation. Moreover, author also performed the principal component analyses of antioxidants accumulation in the different tissues, stages, treatments, and genotypes. The paper is well written, and the conclusions presented are sound and supported by the experiments. I have a few comments that I ask the author to take into consideration in their revision.

1.     Abstract:

Line 21: ‘FT’-write the full name before the abbreviation.

2.     Introduction:

Lines 40-41: please specify the EC numbers of APX, SOD, GPX, and CAT.

3.     Results:

Line 86: it is recommendable to place the section of ‘2.1 Principal component analyses of antioxidants’ at the end of Results.

Line 100: please add the full names of the abbreviation from the figure into the figure legend.

Line 111: delete ‘concentrations’

Lines 182-183: give a reference for dividing the ‘moderate positive (r0.4)and strong positive (r0.6.

4.     Discussion:

Lines 199-200: the conclusion is not entirely correct. Under CLT treatment, the levels of NADP+ were higher in the crown after DEA and REA.

5.     Materials and Methods:

Line 333: delete ‘deacclimation’ and ‘reacclimation’.

Lines 355-356: give a reference for the ‘Spearman’s Rank Correlation Coefficient'.

Author Response

The authors would like to thank the reviewers for their constructive comments and suggestions. We have carefully considered the comments and tried our best to address every one of them. Below are provided the changes, carried out according to the comments by Reviewer 2. For clarity, here we refer to the line numbers from the original manuscript.

Line 21: ‘FT’-write the full name before the abbreviation.

Line 21: abbreviation was explained - “freezing tolerance” (FT).

Lines 40-41: please specify the EC numbers of APX, SOD, GPX, and CAT.

Lines 40-41: the EC numbers were provided (“<…> ascorbate peroxidase (APX) (EC 1.11.1.11), superoxide dismutase (SOD) (EC 1.15.1.1), glutathione peroxidase (GPX) (EC 1.11.1.9), and catalase (CAT) (EC 1.11.1.6) <…>”).

Line 86: it is recommendable to place the section of ‘2.1 Principal component analyses of antioxidants’ at the end of Results.

We respectfully acknowledge this suggestion. However, due to the structure of our manuscript, it is our understanding that section “2.1 Principal component analyses of antioxidants” should remain at the beginning of Results, as it provides an overview of the data. The subsequent sections then delve into different layers of data, comparing the differences of specific antioxidant accumulation between the tissues and treatment groups throughout all stages of acclimation. The Results end with the section “2.5. Correlations between antioxidants and freezing tolerance”. Therefore, if section 2.1 was placed after section 2.5, the logical flow of the Results part of the manuscript would be disrupted.

Line 100: please add the full names of the abbreviation from the figure into the figure legend.

Line 100: abbreviation descriptions for “CTRL”, “CA”, “DEA”, “REA”, “CLT” and “PHLT” were included.

Line 111: delete ‘concentrations’

Line 111: the word “concentrations” was deleted.

Lines 182-183: give a reference for dividing the ‘moderate positive (r≥0.4) and ‘strong positive (r≥0.6).

Lines 182-183: reference was provided.

Lines 199-200: the conclusion is not entirely correct. Under CLT treatment, the levels of NADP+ were higher in the crown after DEA and REA.

Lines 199-200: the sentence was corrected – “Moreover, under the PHLT treatment, leaves tended to contain higher levels of NADP+ <…>”.

Line 333: delete ‘deacclimation’ and ‘reacclimation’.

Line 333: “deacclimation” and “reacclimation” were deleted.

Lines 355-356: give a reference for the ‘Spearman’s Rank Correlation Coefficient'.

Lines 355-356: the reference was provided.

Additional changes were made in accordance with Reviewer 2’s suggestions to provide references for the statistical tests used:

Lines 351-352: references for the Shapiro-Wilk, Wilcoxon rank-sum and Kruskal-Wallis H tests were inserted.

Further changes were made to improve the manuscript:

Supplementary material: Figure S1 and Figure S2 were added; a reference was included.

Lines 24-25: “abiotic stress” keyword was added; the keywords were sorted in the alphabetical order.

Line 176: “The correlations” was changed to “Correlations”.

Line 339: the heading “4.3 Antioxidant assays” was formatted to match the style of the remaining headings.

Line 384: two additional supplementary figures were included – “Figure S1: Principal component analysis (PCA) of antioxidants, measured throughout the experiment under constant low-temperature (CLT) and prolonged higher low-temperature (PHLT) treatments, grouped by genotype; Figure S2: LT30 (temperature, under which 30% of plants die) values of six different winter wheat genotypes after cold acclimation (CA), deacclimation (DEA) and reacclimation (REA) under constant low-temperature (CLT) and prolonged higher low-temperature (PHLT) treatments. This data was discussed by Vaitkevičiūtė et al. (2022).”.

Lines 405-547: scientific names of organisms and genes within the References section were italicized. The formatting was fixed, e.g. “H2O2” and “NAD+” were changed to “H2O2” and “NAD+”.

Lines 542-544: the “Pang et al. (2021) reference was removed, as MetaboAnalyst 5.0 was not used for the final statistical analyses presented in this manuscript.

Round 2

Reviewer 1 Report

The authors improved sufficiently the manuscript, and I consider it ready for publication in Plants .